# Compact terahertz harmonic generation in the Reststrahlenband using a graphene-embedded metallic split ring resonator array

Alessandra Di Gaspare [1], Chao Song[2], Chiara Schiattarella[1], Lianhe H. Li [3], Mohammed Salih[3], A. Giles Davies [3], Edmund H. Linfield [3], Jincan Zhang [4], Osman Balci [4], Andrea C. Ferrari [4], Sukhdeep Dhillon[2] & Miriam S. Vitiello [1]✉

Harmonic generation is a result of a strong non-linear interaction between light and matter. It is a key technology for optics, as it allows the conversion of optical signals to higher frequencies. Owing to its intrinsically large and electrically tunable non-linear optical response, graphene has been used for high harmonic generation but, until now, only at frequencies < 2 THz, and with high-power ultrafast table-top lasers or accelerator-based structures. Here, we demonstrate third harmonic generation at 9.63 THz by optically pumping single-layer graphene, coupled to a circular split ring resonator (CSRR) array, with a 3.21 THz frequency quantum cascade laser (QCL). Combined with the high graphene nonlinearity, the mode confinement provided by the optically-pumped CSRR enhances the pump power density as well as that at the third harmonic, permitting harmonic generation. This approach enables potential access to a frequency range (6-12 THz) where compact sources remain difficult to obtain, owing to the Reststrahlenband of typical III-V semiconductors.

The development of compact technologies for the generation of light across the mid-infrared (MIR) (2.6–24.2 μm)[1,2] and terahertz (THz) (60–250 μm)[3–5] regions of the electromagnetic spectrum has unlocked a plethora of new imaging and sensing methodologies[5]. These enabled the study of fundamental light-matter interactions across the physical[6], chemical[7] and biological[8,9] sciences. The key disruptor has been the quantum cascade laser (QCL)[10], which progressed from being a laboratory curiosity to become an essential, practical and compact photonic source for a broad range of applications[11,12].

This expansion of applications has, however, highlighted a significant gap in technological capability in the spectral region between the MIR and THz, i.e. 25–60 μm (12–5 THz), often referred to as the far-infrared (FIR). Unlike the neighboring MIR and THz regions, the FIR lacks a practical semiconductor laser technology, owing to parasitic optical phonon absorption – the *Reststrahlenband* – in the constituent III-V semiconductors used to fabricate QCLs[10]. This is unfortunate, since there is a large number of applications that could be addressed by a compact FIR source, ranging from sensing complex hydrocarbons in the petroleum-industry[13], to probing the protein functions in amino acids[14]. FIR sources are also an enabling technology for near-field microscopy in the solid state (phononics, plasmonics)[15], and for quantum optics, e.g. in the manipulation of Rydberg atoms in quantum computation architectures[16].

There is no practical, spectrally narrowband, technology to access the FIR at present, to the best of our knowledge. Current techniques are either difficult to use in real time, in situ applications, or they suffer from poor performance. Thermal sources used in Fourier transform infra-red (FTIR) spectroscopy are the most common approach, but they are inherently incoherent, weak and spectrally broad. Alternative commercial systems include time-domain spectroscopy (TDS), in

[1]NEST, CNR-NANO and Scuola Normale Superiore, 56127 Pisa, Italy. [2]Laboratoire de Physique de l'Ecole Normale Supérieure, ENS, Université PSL, CNRS, Sorbonne Université, Université de Paris Cité, Paris, France. [3]School of Electronic and Electrical Engineering, University of Leeds, Leeds LS2 9JT, UK. [4]Cambridge Graphene Centre, University of Cambridge, Cambridge CB3 0FA, UK. ✉e-mail: miriam.vitiello@sns.it

which III-V photoconductive switches[17], or spintronic heterostructures[18,19], generate coherent radiation pulses through femtosecond excitation, but these are also spectrally broad. Furthermore, TDS systems are not compact and emit most power over the -0.2–4 THz range. Spintronic emitters can be also employed to generate broad-band THz light in a pumped optical scheme via up-conversion[20].The few existing narrowband sources, other than facility-based free electron lasers, rely on difference frequency generation in exotic gas lasers (e.g. $^{15}NH_3$)[21], which are cumbersome and lack stability[21]. Present research is focused on identifying new III-V semiconductor systems for spectrally narrowband and powerful laser sources, including SiGe[22] and ZnO[23], for the THz range[24]. However, due to the complexity of their impurity and phonon spectra (-9 THz), it is challenging to reach the 25–50 μm (6–12 THz) FIR spectral window.

On the other hand, nonlinearities in layered materials (LMs) are promising for the generation of THz or FIR light[25,26], due to their unique optical and electronic properties, distinct from those of bulk materials, including III-V semiconductors. Single layer graphene (SLG) has a third order nonlinearity $\chi^{(3)} \sim 10^{-9}$ m$^2$/V$^2$, i.e. -15 orders of magnitude higher than typical 'THz' materials[25,27,28]. Combined with SLG's ultrafast dynamics[29], this has permitted THz up-conversion - harmonic generation - where light can be converted from a low to a higher photon energy at harmonics of the former. High harmonic generation at room temperature was reported using moderate fields, up to the 7th harmonic at frequencies ≤2.2 THz with an optical pump at 0.3 THz[26,30–32]. In contrast to SLG that only possesses a third order nonlinearity, THz second harmonic generation can be achieved using spintronic heterostructures[20]. However, all these demonstrations, limited to low THz frequencies[33], require entire facilities or large table-top laser systems as THz pump sources, thus limiting their applicability.

Here, we combine resonant nonlinear optics in SLG and strong optical confinement to demonstrate FIR emission at 9.63 THz in a compact geometry. This is performed by optically pumping SLG coupled to a circular split ring resonator (CSRR) with a high-power semiconductor laser – a QCL – emitting at a central frequency of 3.21 THz (see Fig. 1a)[34]. The mode confinement provided by the CSRR, combined with the SLG nonlinearity, is essential to enhance the pump power density from the THz QCL, allowing third harmonic generation (THG) in SLG in a frequency range (>3 THz) unexplored so far, where compact sources do not exist. As well as generating light in the *Reststrahlenband*, our results show that THz QCLs can be used to explore high field THz physics, where a band structure-by-design approach can be used to target narrowband resonances in, e.g., plasmons, magnons and phonons.

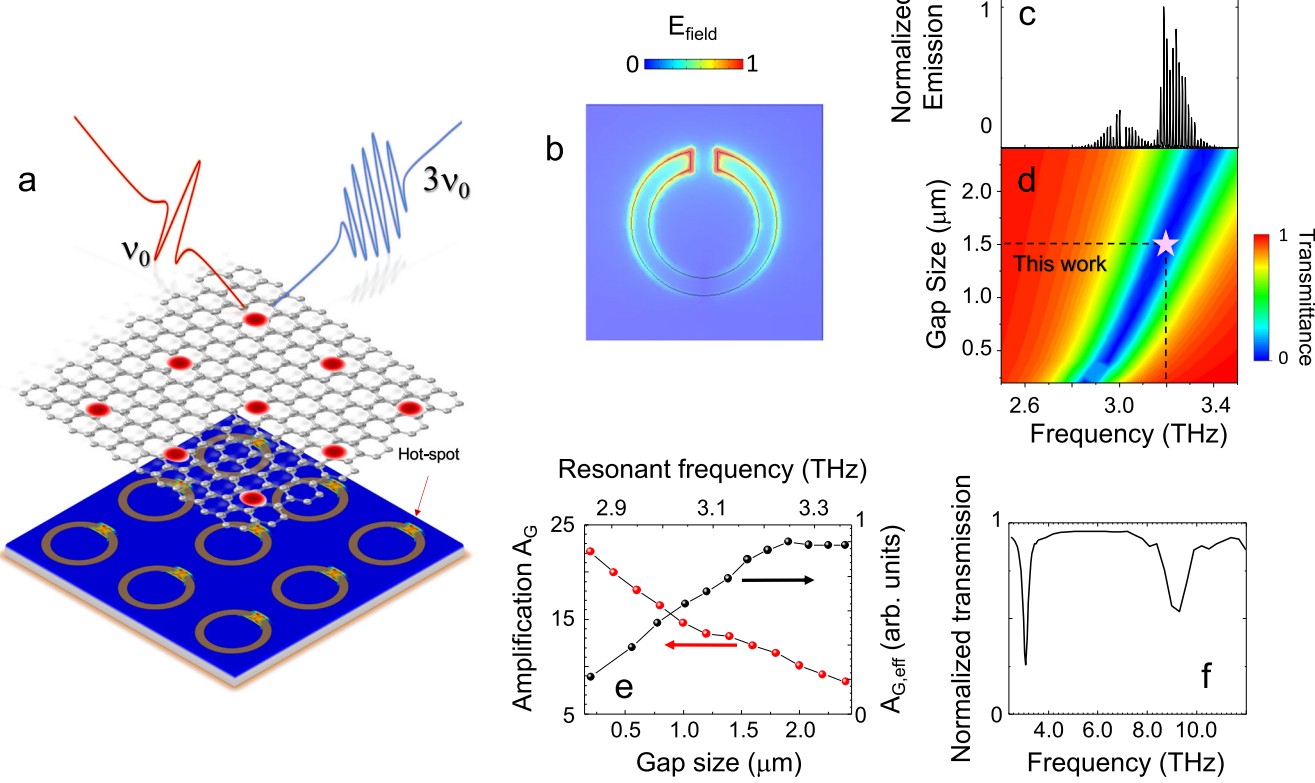

**Fig. 1 | SLG coupled to a circular split ring resonator (CSRR), resonant at 3.21 THz. a** Device concept: CSRR array resonant at a selected THz frequency $\nu_0$, covered with single layer graphene (SLG) for frequency up-conversion. Third harmonic generation (THG) is a result of the strong field enhancement in the gap of the SLG-CSRR. **b** Two-dimensional (2d) map of the optical electric field amplitude, calculated by the finite element method (FEM) for the bare CSRR, at $\nu_0 = 3.21$ THz. **c** Fourier transform infrared (FTIR) emission spectrum of the quantum cascade laser (QCL) pump source. The SLG CSRR is designed to be resonant with the central frequency of the QCL optical bandwidth. **d** 2d maps of transmittance, calculated as the square amplitude of the S12 coefficient (see Methods) extracted from simulations, as a function of gap size, and corresponding resonance frequency, obtained from simulations. The intersection of the dashed lines and the star symbol marks the experimental condition adopted here. **e** Field enhancement of CSRR ($A_G$, amplification gap, red circles), and normalized amplification, $A_{G,eff}$ (black circles, normalizing $A_G$ for the effective extension of the enhancement region), plotted as a function of the split gap size (bottom axis) and of the corresponding resonance frequency (top axis). The field enhancement is extracted by running simulations with the gap size as the only parametric value, while keeping the CSRR geometry (ring radius, $r = 3.6$ μm, ring width, $w = 1$ μm, and period $p = 14.5$ μm) fixed. **f** Simulated transmission spectra, extracted from the FEM simulations, for the resonator geometry in **d**, over the 2–12 THz frequency range, showing the fundamental third harmonic modes of interest for THG.

## Results

### SLG-embedded metallic split ring resonator array

To overcome the challenges imposed by the small Drude weight of the SLG conductivity at ~3 THz[35,36], which limits the conversion efficiency for harmonic generation, we embed SLG (see Methods) in a resonant structure, comprising of an array of micrometric C-shaped ring resonators (CSRR). This shifts and enhances the SLG Drude conductivity at the QCL pump frequency, which, in turn, maximizes the light–matter interaction region, and permits THG to be observed. The field enhancement in the split gap of each CSRR[37,38] is one order of magnitude larger than the absorption in an un-patterned film (i.e. no resonator)[39,40]. In the un-patterned SLG, the nonlinear response arises from the SLG direct absorption[26]. The introduction of a metamaterial-based structure is an effective solution to concentrate the optical field in the SLG[30,32], thus enhancing the SLG absorption. In the CSRR case, this leads to the formation of a hot-spot in the split gap, which determines an out-of-equilibrium hot electron state[29,41], as confirmed both by electromagnetic simulations and near-field nano-imaging (see below).

The two-dimensional (2d) resonator array comprises a set of Au, CSRRs, patterned onto a SiO$_2$/Si substrate (Fig. 1b). To design the CSRR, we first set the unit cell geometrical parameters, e.g. ring diameter, width and gap size, to match the desired resonance for frequency up-conversion (see Supplementary Note 1) at the central frequency (3.21 THz) of the pump source, e.g. the THz frequency of the QCL $\nu_O$ (Fig. 1c). The electromagnetic response, extracted by finite element method (FEM) simulations (Comsol Multiphysics), exhibits a tunable-by-design absorption band with a specific resonant frequency, and a field-enhancement (Fig.1b) in the split gap. Hence, the CSRR design offers the capability to exploit the split gap for SLG embedding. This allows us to concentrate the peak electric field in a portion of the active material where no screening or absorption from the metallic counterpart of the array dominates. This is in contrast with simpler grating arrays[32], where the electromagnetic coupling efficiency drops as 1/f[33], hence being less suitable for the high THz frequencies aimed in the present work.

The electromagnetic coupling mechanism between the CSRR and SLG is a combination of capacitive coupling due to the CSRR gap and inductive coupling due to its ring shape. The magnetic inductive current helps building the electric field in the split gap, thus enhancing the field for THG.

Figure 1d shows the frequency dependent transmittace map of for different split gap sizes. The resonance frequency is affected by the gap size, revealing the interplay between the parameters needed to set the optimal geometry[37]. The field enhancement of the CSRR ($A_G$), defined as the ratio between the average field in the resonator gap and that in an equally extended surface outside the gap region, i.e at the ring center, is shown in Fig. 1e. The field strength in the gap is higher in resonators with smaller sized gaps. However, in smaller gap resonators, the effective area of the pumped active film and the size of the hotspot region is reduced[42]. Thus, we define an effective field enhancement $A_{G,eff}$ by normalizing $A_G$ to the effective hot-spot area (black dots in Fig. 1e). The optimal combination of gap size, enhancement factor and resonance frequency, is shown for a 1.5-μm split-gap. This achieves a maximum enhancement factor ~13, corresponding to a calculated quality factor Q ~ 13 for a CSRR designed to be resonant at 3.21 THz. In our experiment, the pump signal is absorbed by the SLG-embedded resonator, driving, and enhancing, its non-linear response. The pump is then up-converted for generation at 3$\nu_0$. The CSRR enhances the pump field in the up-conversion of the incoming field, as well as contributing to the emitted third harmonic signal, since CSRR arrays have resonant modes at the third harmonic frequency (Fig. 1f), hence boosting the overall THG efficiency.

To contain the optical screening of the metallic CSRR array and the consequent resonance bleaching if the SLG film covers the entire ring area (see Supplementary Note 1), we integrate SLG only in the CSRR split gap (see Fig. 2a). The SLG-CSRR transmittance, measured by FTIR spectroscopy in vacuum (Fig. 2b), shows a resonant absorption dip slightly weaker (~22%) and blue-shifted (0.20 THz) than that measured in the bare CSRR in an identical experimental configuration.

Comparison with the simulated design reveals a good agreement with the calculated full-width-at-half maximum (FWHM) of the absorption dip in the bare CSRR, and a broader dip in the SLG-CSRR, corresponding to $Q_{bare} = 10.5$ for the bare CSSR and $Q_{gra} = 4.5$ for the SLG-CSRR (see Fig. 2c). If compared to linear grating dipolar resonators, the ring design permits a higher Q-factor (See Supplementary Note 4).

A more accurate model that accounts for the SLG integration in the CSRR gap requires consideration of the SLG complex optical constants. Within the Drude model[35], the SLG intraband conductivity is a doping- and scattering time- dependent function:

$$\sigma_{intra}(\nu) = \frac{-iD_0}{\pi} \frac{1}{(2\pi\nu + i\Gamma_0)} \tag{1}$$

where $D_0 = E_F e^2/\hbar^2$ is the linear Drude weight, $e$ is the electron charge, $\hbar$ is the reduced Planck constant, $\Gamma_0 = \tau_0^{-1} = ev_F^2/E_F\mu$ is the scattering rate, $E_F$ the Fermi energy, $v_F$ the Fermi velocity, and $\mu$ is the carrier mobility. By setting $E_F = 250$ meV and $\tau_0$ 23.4 fs (see Supplementary Notes 2–3), we reproduce the experimental blue-shift and broadening of the absorption dip in Fig. 2c.

To map the optical field enhancement in the resonator gap induced by THz photoexcitation, we perform near-field nanoscopy experiments on an individual SLG-CSRR, employing a THz QCL emitting 19 equally spaced lasing lines (central frequency $\nu_C = 2.94$ THz, frequency spacing $\Delta\nu = 38.5$ GHz, Fig. 2d). We employ a detector-less technique (Fig. 2e), with the QCL acting simultaneously as both THz source and phase-sensitive detector in a self-mixing (SM) interferometric scheme[43,44]. (see Methods). The coherent superposition of the reinjected THz field with the QCL produces a perturbation of the laser voltage ΔV that depends on both amplitude and phase of the THz field scattered by the tip (see Methods). We acquire the sample topography (Fig. 2f) simultaneously with the total back-scattered signal (Fig. 2g). The near-field map of the third-order SM amplitude signal ΔV$_3$ (Fig. 2e) has an optical contrast modulation that reveals the different reflectivity of CSRR array elements, metal ring, SLG film and Si/SiO$_2$ substrate. The near field signal from the SLG surface outside of the split gap region contrasts with the corresponding signal from the insulating Si/SiO$_2$ substrate, being 2.1 times larger.

This is expected from the higher optical contrast/local reflectivity of SLG when compared to the Si/SiO$_2$ substrate. Optical amplification, with an enhancement factor ~3.8, happens only in the resonator split gap, as highlighted in Fig. 2g. This is a factor ~3.4 lower than that computed via electromagnetic simulations (Fig. 1e). The discrepancy arises from the optical bandwidth of the QCL comb used for the near-field, with only one mode overlapping the SRR bandwidth. By averaging the amplification factors extracted from the simulation at each comb tooth frequency, and then normalizing the result by the corresponding intensity, we obtain an average enhancement ~3.31, in agreement with experiments.

### Nonlinear optical response

The hot-electron distribution in SLG, expected to induce a non-linear response[45], is affected both by the local field enhancement in the split gap and by the overall absorption of the CSRR array. Although the implementation of high (~100) $Q_s$ could be the most convenient pathway to promote nonlinear effects in SLG at high (~50 kV/cm) fields[39], the complex interplay between non-linear response and shape of the absorption dip can lead to a counterintuitive outcome. Indeed,

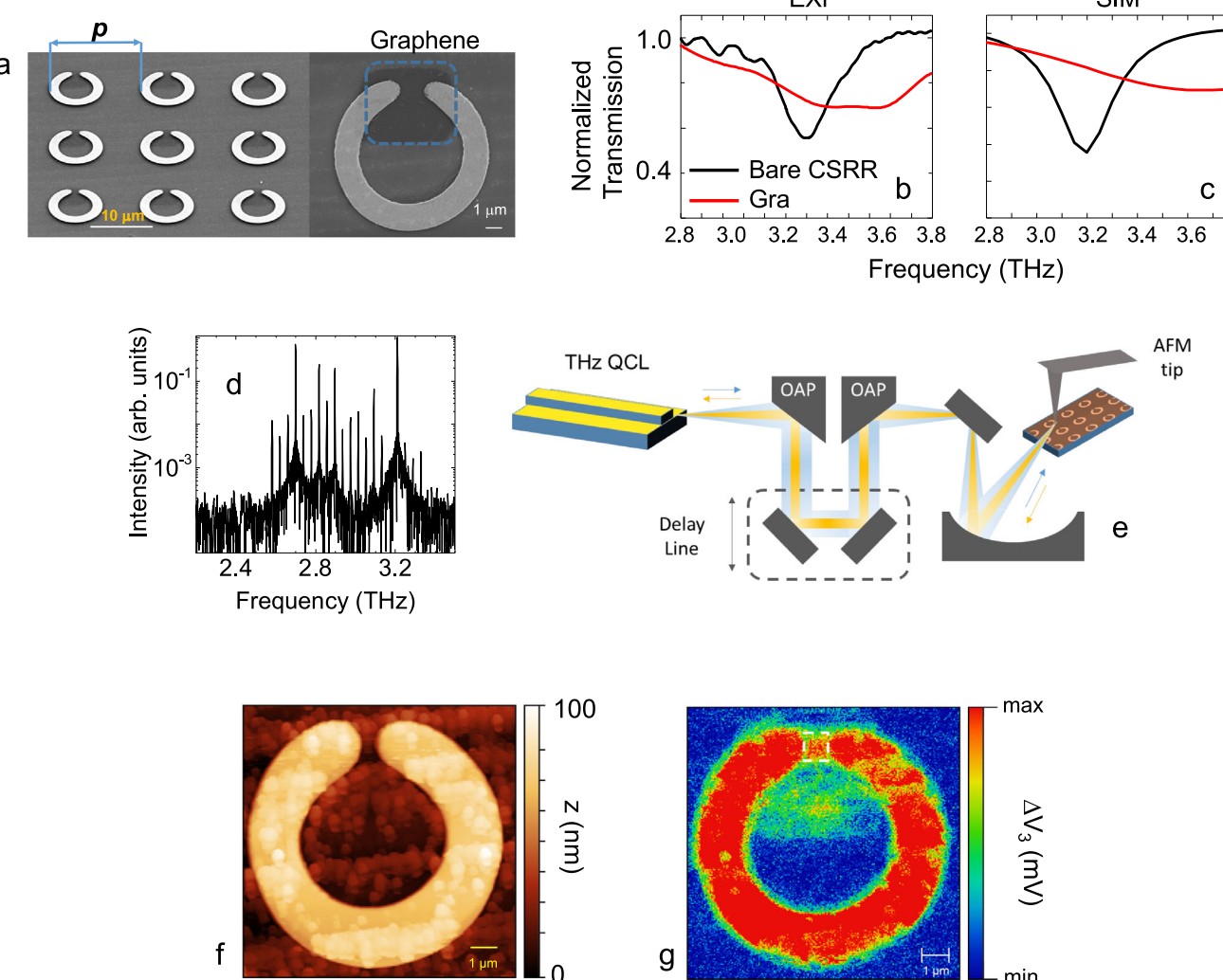

**Fig. 2 | Field enhancement in the SLG CSRR gap. a** Scanning electron microscope (SEM) image of optimized CSRR array, with period *p*, patterned with SLG that only covers the split gap area (blue dashed area on the right), where the field enhancement is maximized. The dotted blue area marks SLG. **b** Experimental and (**c**) simulated transmission of bare CSRR (black) and SLG-CSRR (red) arrays. The transmission curves are normalized by the reference sample trace, acquired on a portion of a bare SiO$_2$/Si substrate. **d** FTIR emission spectrum of QCL frequency comb with spectral bandwidth~0.7 THz centered ~2.94 THz. **e** Schematic near-field scattering detector-less nanoscope, employing a THz QCL frequency comb as source and detector simultaneously. The self-mixing signal, collected to retrieve the near field maps, arises from re-injection of the backscattered beam from the atomic force microscopy (AFM) tip of the scattering type scanning near field optical microscope (s-SNOM) into the laser facet. A delay-line is used to control the length *L* of the optical path and tune the optical feedback. OAP: off axis parabolic mirror. **f** Topographic maps of a single SLG-CSRR. **g** Near-field map of third-order self-mixing signal $\Delta V_3$ of the CSRR in panel a (right side). The white dashed area highlights the field enhancement in the slit gap. The near-field map is collected with the radiation approaching the sample at a 45° incident angle. The sample is therefore excited with two equally contributing *s*- and *p*- terms with respect to the CSRR plane, under the hypothesis of a purely collimated beam. However, the focusing of the beam onto the SNOM tip also implies a spreading of the probed incident wavevectors *k* (therefore of the polarizations, orthogonal to *k*), dictated by the beam radius and the numerical aperture of the focusing component, i.e. the SNOM parabolic mirror in front of the tip.

the presence of a 3$^{rd}$ order resonance in the CSRR (see Supplemtary Note 4) plays a role in the THG process, affecting its efficiency.

We thus consider the total field enhancement by normalizing the absorption of the CSRR, calculated by considering a Lorentz-like electromagnetic response of the resonator, resembling a damped oscillator at central frequency $v_0$ and nominally different $Q_0$ values (see Fig. 3a), with the maximum field enhancement ($A_G$, see Fig. 1e).

The presence of a 3$^{rd}$ order resonant mode having a quality factor $Q_3$ (see Supplementary Note 4) is reflected in the field enhancement, which also includes a 3$^{rd}$ order harmonic term. We select a range of $Q$ defined by the two extreme cases, i.e. an isolated CSRR with $Q$ set only by geometry, and the packed array case, where the radiative losses from ring cross-talking induces a significant broadening of the

resonance. This results in $Q$ from 75 (ideal, zero losses) to 6 (radiative losses from inter-lattice cross talk, see Supplementary Note 4). $Q$ and $Q_3$ have a similar dependence from the period $p$ of the CSRR array (Fig. 3b), with $Q_3$ a factor ~2.3–1.2 larger than $Q$ across our range of periods. $A_G$ of the 3$^{rd}$ order mode is < 25% weaker than that of the fundamental mode.

Here, the hot-electron dynamics in the SLG Dirac-like system affects its optical response to external photoexcitation. The effective electric field in the gap, i.e. on the SLG, $E_0(\nu)$, Fig. 3c, comprises the external optical beam (inset Fig. 3c), a quasi-monochromatic, linearly polarized, radiation field at normal incidence. The beam delivers 1 W optical power focused on a 0.7 mm diameter circular spot. This is amplified by the field enhancement shown in Fig. 3a, defined as the SLG CSRR absorption normalized by $A_G$. $E_0(\nu_0)$ on SLG-CSRR is over one

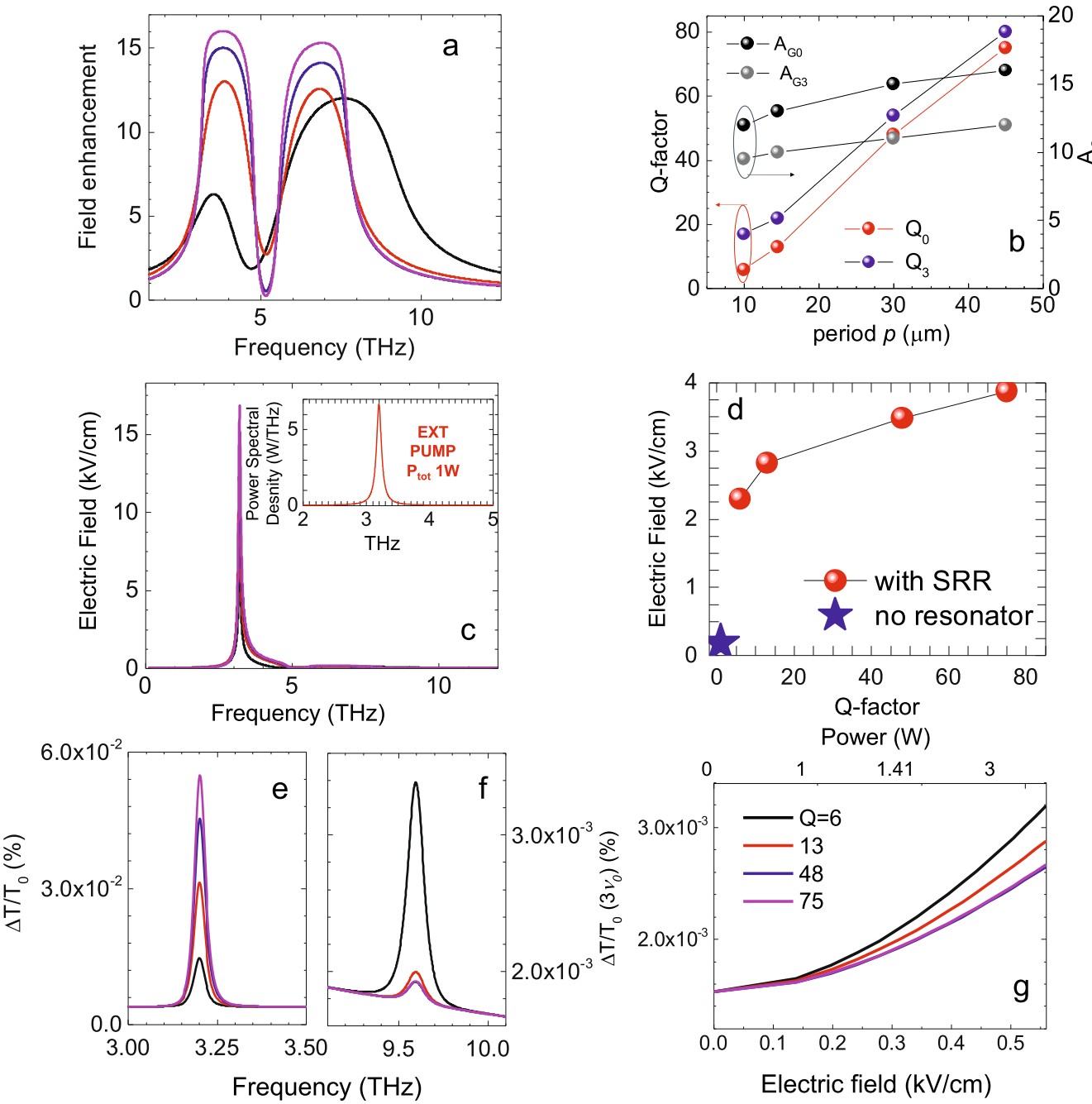

**Fig. 3 | Simulated THG efficiency in THz QCL pumped SLG-CSRR. a** Field enhancement in SLG-CSRR as a function of frequency, for different resonator array quality factors ($Q$), representing the SLG-CSRR absorption normalized by $A_G$. Different colors correspond to $Q = 75$ (purple), 48 (blue), 13 (red), 6 (black). **b** $A_{G0}$ and $A_{G3}$ (right axis) calculated as the ratio between average optical field in- and outside of the SRR gap, extracted from the FEM simulation for the fundamental (black) and third harmonic modes (gray), shown alongside the respective $Q$ (left axis) labeled as $Q_0$ (red) and $Q_3$ (blue), as a function of SRR array period. **c** Electric field on SLG-CSRR as a function of frequency, obtained from the convolution of the field enhancement of a) and the optical beam intensity in the inset of **c**, assuming 1 W total emission power focused on a 700 μm diameter circular spot, whose frequency distribution is shown in the inset, for different $Q$. The color code is the same as in panel **a**. Inset: **d** Electric field, calculated at the fundamental frequency (red dots) in SLG-CSRR and SLG (no SRR, blue star), as a function of $Q$, corresponding to the $p$ values of **b**. **e, f** Normalized transmittance variation as a function of frequency, $\Delta T/T_0$ (ν), in the pumped SLG-CSRR, assuming an incident effective electric field as in **c**, following Eq. (3), at the fundamental QCL frequency $\nu_0$(d), and at $3\nu_0$ (**e**), for different $Q$ values (same code as in **a**). **g** THG efficiency, defined as $\Delta T/T_0$ ($3\nu_0$) in the SLG-CSRR, as a function of impinging electric field (bottom axis) and total power (top axis), assuming an incident beam diameter of 700 μm, calculated at different $Q$ (same color code is the same as in **a**).

order of magnitude larger than that calculated on SLG only (Fig. 3d). The corresponding $Q$, calculated at $p = 45$ μm, is ~83% larger than that at $p = 10$ μm, as shown in Fig. 3b (red dots).

Under steady state excitation[46], where the optical beam is either a continuous wave or has a pulse duration much longer than the SLG cooling time (>2.5 ps)[29,47,48], the impinging electric field induces a

polarization current in the SLG, oscillating in the same direction of the incoming field (see Supplementary Note 5) at the frequency of the incident wave[27,28,49]. The SLG non-linear response relies on the hot carrier population driven by efficient intraband absorption. It can be modeled by using the SLG Drude response, taking into account the heating/cooling dynamics of the carrier population, and the time

profile of the excitation beam[25,30,42]. However, at higher optical field intensity (≥10 kV/cm), the presence of inherent nonlinear components in the optical response also suggests field-driven nonlinearities beyond thermal effects, accounted by the Kerr term[42] (see Supplementary Note 5). In doped SLG, the absorption of an intense, high-fluence THz frequency beam, may lead to overheating, a phenomenon known as "hot-carrier multiplication", ascribed to the fast thermalization of the free carriers, accompanied by a reduction of the absorption coefficient[31]. Although the fluence is significantly higher in the present case of quasi-continuous wave (CW) source (pulse duration 1 μs, fluence > 350 μJcm$^{-2}$), the peak electric field is over one order of magnitude lower than that achievable in pulsed sources. The thermodynamic equation for thermalization dynamics (see Supplementary Note 5), leads to an electronic temperature significantly lower than that retrieved in the case of pulsed sources[31]. Hence, overheating effects can be neglected in the present case.

The non-linear optical response can then be calculated by following the method of refs. 30,31,50:

$$\sigma_{3,tot}(\nu) = \sigma_{intra}(\nu) + |E_0(\nu)|^2 \times \sigma_3(\nu) \qquad (2)$$

with $\sigma_3(\nu)$ the non-linear Kerr optical conductivity. Perturbative models[51–53] based on microscopic scattering formalisms encoding relaxation times for both intraband and interband absorption are valid under the assumption of normal incidence, and for a finite-time pulse excitation. Both conditions do not apply to our experimental configuration. Therefore, we use the semi-empirical model of refs. 25,26,30, working well in predicting the THG conversion efficiencies, by calculating the hot-electrons optical conductivity, under steady excitation conditions, to mimic the THz quasi-CW beam.

In Eq. (2), the 3$^{rd}$ order conductivity is then calculated assuming a hot-electron temperature ($T_e$) dependence of the Drude coefficients, and including the contribution of the plasmonic frequency dependence, accounted for by introducing the effective frequency $\nu_{eff} = (\nu^2 - \nu_0^2)/\nu$. The linear Drude weight ($D_0$) and scattering rate ($\Gamma_0$) in Eq. (1), are valid for T = 0. At T ≠ 0, the smearing-out of the Fermi-Dirac distribution could open up alternative channels for hot-carriers relaxation, e.g. interband transitions, and the model would need to take into account different relaxation paths and their contribution to the conductivity. These effects are negligible for high-doping and low energy photons, i.e. under our conditions where: $E_F \gg \hbar\omega$ and $E_F \gg k_B T_e$[31,46]. The purely Drude-like and hot-electron terms are an excellent approximation for the THz intraband absorption, and they can be extended to the T ≠ 0 case. When the system nonlinear response is stimulated, the detected optical signal deviates from the expected linear signal. The laser beam illuminating the SRR array is transmitted to the detector after interacting with the SRR sample. The nonlinear response induces a transmittance variation $\Delta T/T_0 = |T_3 - T_0|/T_0$, whose frequency dependence (Fig. 3e) can be derived from the linear ($T_0$), and non-linear ($T_3$) transmittances as:

$$[T_0, T_3(\nu)] = \frac{4n_{sub}}{Z_0^2 \left| \frac{n_{sub}}{Z_0} + [\sigma_0, \sigma_{3,tot}(\nu)] \right|^2} \qquad (3)$$

where $Z_0$ is the vacuum impedance. Such parameter also represents the THG efficiency, defined as the signal intensity variation at $3\nu_0$ over the signal intensity at the fundamental (linear) frequency[26,30].

We observe an overall increase of the optical transmission (Fig. 3e, f), resembling optical absorption bleaching, or saturation, reported in SLG[54–56] due to photo-induced conductivity[28]. The calculated $\Delta T/T_O$ in SLG-CSRR at $\nu = \nu_0$ increases for higher $Q$ (Fig. 3e). Conversely, $\Delta T/T_O$ at $\nu = 3\nu_0$, decreases at high $Q$ (Fig. 3f), as an effect of the larger $\sigma_{3,tot}$ in SRRs having broader resonances. This reflects in a THG efficiency larger than that expected in narrower $\Delta T/T_O$ ($\nu_0$)

(Fig. 3f). Plotting the THG efficiency, $\Delta T/T_O$ ($3\nu_0$)[26,30] (Fig. 3g), as a function of electric field, one can see that the THG efficiency increases at high electric fields and in SRRs having low/moderate $Q_s$. As the input electric field and $Q$ simultaneously increase, only the term at the fundamental frequency $\Delta T/T_O$ ($\nu_0$) becomes more pronounced if compared with the 3$^{rd}$ harmonic one $\Delta T/T_O$ ($3\nu_0$), hence quenching any possible THG effect.

## Third harmonic generation in optically pumped SLG-CSRR arrays

To demonstrate THG in the SLG-CSRR, we fabricate an array of SLG-CSRR with $p = 14.5$ μm, maintaining the geometrical parameters of individual SRRs as in Fig. 1. The SLG-CSRR is initially mounted in the internal unit of a FTIR under vacuum (Bruker, Vertex 80 v) illuminated, in transmission mode, by the internal source of the FTIR spectrometer to measure the transmission without and with the incident QCL beam (see Supplementary Note 6). In this case, the dip in the transmittance is weaker (~10%), in overall agreement with the expected non-linear increase of SLG transmittance under intense optical excitation, also depicted in Fig. 3e.

The SLG-CSRR array is then optically pumped with a THz frequency QCL, delivering 2.5 W peak optical power, driven in pulsed mode (pulse width = 1 μs), and focused on a 700 μm diameter circular spot on the SLG-CSRR gap (Fig. 4a). To isolate the up-converted signal, we position a ~6 THz high-pass Ta-filter (Crystan limited, see Supplementary Note 7) along the optical path in front of the window of a Si bolometer (IR Lab), and detect the signal emitted by the optically pumped SLG-CSRR with a lock-in amplifier, referenced to the same signal used to amplitude-modulate the QCL.

We initially measure the signal detected by the bolometer with a set of Ta-filters having different thicknesses (Fig. 4b). The resulting data, Fig. 4b, shown after normalizing the signal by the transmission spectrum of each filter, are fitted by a 3$^{rd}$-order polynomial function, confirming the signal is a 3$^{rd}$ order process, as expected from the calculated $\Delta T/T_O$ ($3\nu_0$) of Fig. 3g and Eqs. (2) and (3).

Then, we acquire the signal emitted by the SLG-CSRR, in step-scan mode, filtering the QCL, with the 2-mm-thick high bandpass Ta filter (see Supplementary Note 7), first orienting the SLG-CSSR along the polarization axis of the QCL, then perpendicularly. A peak at ~9.63 THz, the signature of THG, at the 3$^{rd}$ order harmonic of the pump beam (Fig. 4c, top black curve) is only measured when the polarization of the pump beam coincides with the dipolar orientation of the CSRR (Fig. 4c, red curve). If the polarization is in the orthogonal orientation, no signal is visible (Fig. 4c, gray curve). Furthermore, it is not observed, under identical polarization conditions, when an identical metallic split ring is patterned on the substrate (Fig. 4c, green curve), unambiguously proving its origin. In the investigated frequency range, any FIR photons are absorbed by *Reststrahlenband*, hence no THG can arise from the QCL.

## Discussion

The THG signal is ~10$^5$ less intense than the most intense mode of the pumping QCL, in agreement with the conversion efficiency of $\Delta T/T_O$ ($3\nu_0$) ~ 2.42 × 10$^{-5}$ calculated for our experimental configuration (blue curve, Fig. 4c) from the ratio between the detector signals retrieved with and without the ~ ~6 THz cutoff filter. Considering the flat spectral response of the Si-bolometer in the 3–10 THz range, we estimate a THG signal around ~7 μW at the QCL peak emission. Such a conversion efficiency is over two order of magnitude smaller that than reported in refs. 26,30, owing to the steep reduction of the optical conductivity in the sub-THz if compared to >3 THz range[35,42], and especially to the significantly higher field strength, even when accounting for the field enhancement provided by the CSRR. In a bare SLG film, the expected THG efficiency, in the same experimental configuration,

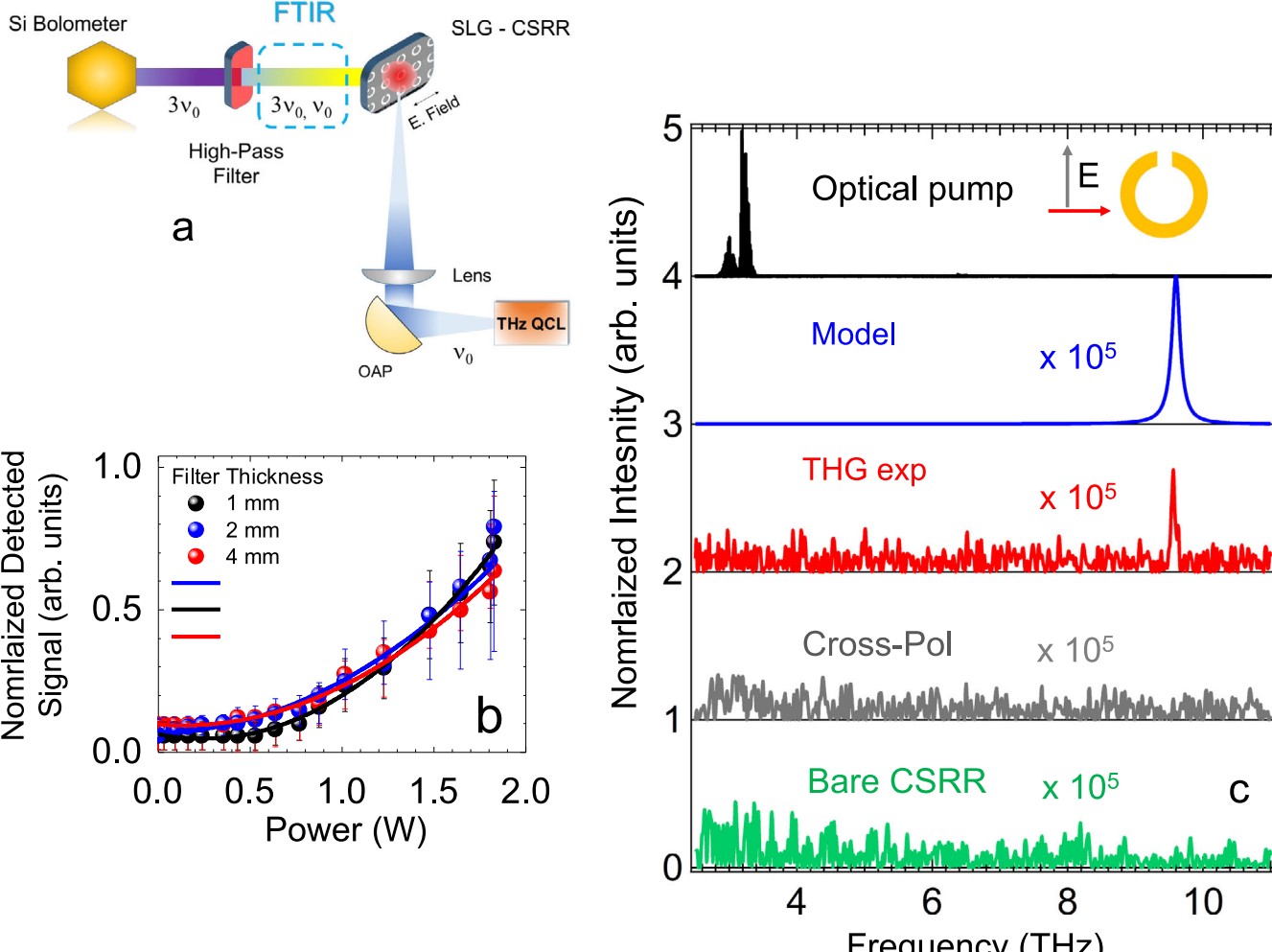

**Fig. 4 | THG at 9.63 THz in optically pumped SLG-CSRR. a** Schematic of the experiment. Light from a single-plasmon QCL, emitting 2.5 W in the free space, is focused on the SLG-CSRR array. The THG spectral signal is isolated after filtering the THz QCL with a ~6 THz high bandpass Ta-filter. **b** Signal measured by the Si-bolometer with the experimental arrangement in **a**, while keeping the FTIR moving mirror fixed, as a function of QCL power, after filtering the QCL with Ta-filters of different thickness: 1 mm (black circles, high-pass transmittance ~65%), 2 mm (blue, ~59%), 4 mm (red, ~55%). The detected signal is normalized by the transmittance of each filter. The QCL driver is amplitude-modulated with a 317 Hz TTL signal, while acquiring the signal detected by a He-cooled Si-bolometer via a lock-in amplifier. The error bar on the y-axis represents the normalized noise level affecting each measure. The black, blue and red lines represent the fit functions $y = y_0 + B \times (x - x_0)^3$, where $y_0$, $x_0$ and $B$ are free running parameters. The fit procedure converges in the 3 cases with correlation factors $R > 0.9$. **c** From top to bottom; black line: normalized emission spectra of THz QCL acquired at 15 K in rapid scan mode, with spectral resolution ~0.075 cm$^{-1}$ and aperture size ~1 mm, to prevent FTIR detector saturation (y-axis offset +4, black). Blue line: THG efficiency

for an input power $P_{exp} = 1.8$ W, accounting for reflection losses of both cryostat and FTIR optical windows, and a SLG-CSRR $Q_{exp} = 10$ (from Fig. 2b). Red, grey lines: emission spectrum measured on the optically pumped SLG-CSRR, after filtering the QCL pump with the Ta-filter, measured in step-scan mode with a spectral resolution ~1 cm$^{-1}$ and aperture size ~5 mm, with the SRR array oriented parallel (red) and perpendicular (gray) to the polarization axis of the QCL, as shown in the inset by the red and gray arrows, respectively. Green Line: Emission from the optically pumped bare CSRR array sample with the SRR array oriented parallel to the polarization axis of the QCL. For comparison with the top panel, all the experimental (red, gray and green) and theoretical (blue) curves in the panel are normalized by a factor 2.42 × 10$^{-5}$, accounting for the expected THG efficiency. The experimental THG is 0.95 × 10$^{-5}$, determined after accounting for the amplitudes from the different acquisition conditions (gain conversion of lock-in amplifier, FTIR aperture diameter). The inset in panel **c** shows the linear polarization directions employed to acquire the THG emissions: red curve (red arrow, polarization active direction) and gray curve (gray arrow, cross polarization of pump beam with respect to the CSRR array orientation).

---

would be ~0.22 × 10$^{-5}$, corresponding to 0.65 µW, hence below the detection limit.

No signal at the second harmonic (SH) frequency is retrieved. A SH generation signal would rely on a nonzero second-order nonlinearity ($\chi^{(2)} \neq 0$), prohibited by the centro-symmetrical nature of the hexagonal SLG structure.

An anisotropic SH signal could in principle arise in SLG on SiO$_2$[57,58] if the crystal orientation of SLG hexagonal structure with respect to the substrate crystal fulfills the conditions for dipolar symmetry breaking. However, in our polycrystalline SLG, comprising single crystalline domains of few microns size and random crystalline orientation, the

latter condition is not satisfied. Although the introduction of non centrosymmetric resonator arrays could activate/enhance the even-order nonlinearities[59,60], the efficiency of this effect is likely too weak in the proposed system[32]. In ref. 60, the reported SH generation in CSRRs relies a combination of impact ionization and non-linear Thomson scattering, to multiplicate and accelerate the free carriers and establish a significant out-of-plane Lorentz force[60]. While the field enhancement from the asymmetric SRR ($A_G$ ~ 15)[60] is in broad agreement with the value of our design ($A_G$ ~ 13), SH generation was observed with a tens of kV/cm pump field in a FEL, to drive the carrier to the impact ionization. Conversely, we here adopt fields ≲1 kV/cm. Although impact ionization

is possible in SLG[61], the carrier density (≥$10^{18}$ cm$^{-3}$)/field-driven velocity (field ≥25 kV/cm) thresholds are orders of magnitude higher than those adopted in our experimental condition. Furthermore, in our CSRR structure, with SLG patterned only in the split gap, the magnetic coupling enhancement of the CSRR design is spatially separated from the electric field area (see Supplementary Note 8). In our experiments, high harmonic generation results only from electric field enhancement, dominant at both fundamental and at third harmonic frequencies. At the second harmonic, on the other hand, the same amplification is not observed.

In conclusion, we engineered a SLG-CSRR for THG in the FIR range, exploiting the amplified field in the split gap of a CSRR and the intense nonlinear 3$^{rd}$-order optical response in SLG. Emission at 9.63 THz is demonstrated when the SLG embedded in the CSRR is optically pumped with a compact 3.21 THz QCL, a frequency corresponding to the third order up-conversion of the fundamental laser mode. This paves the way for the demonstration of solid-state coherent emission in the FIR spectral range, through on-chip integration of SLG metasurfaces with miniaturized QCLs. Engineering smaller split gap resonators in a double-metal configuration could increase the optical amplification by enhancing field confinement. We used compact sources to target interactions only previously possible in FELs (magnons, phonons, spin, impact ionization[62]). This could lead to QCLs tunble over a phonon resonance, where nonlinearity is enhanced. Our results open the route for a fully integrated approach not possible with FELs or tableto systems. SLG can be patterned over the surface of a THz QCL, whose intracavity power/field would be enough to induce FIR emission by electrical pumping. The same approach can be also adopted with alternative materials, such as topological insulators, that can overcome HHG efficiency limitations arising from SLG optical saturation[63]. This may enable novel applications to be addressed, ranging from high-resolution sensing of complex biomolecules[64] that fall in the *Reststrahlenband* of existing QCLs, high field THz physics, and near-field s-SNOM for quantum nanoscopy in the 25–60 μm range, where many plasmonic, phonon and magnetic phenomena occur.

## Methods

### SLG synthesis and transfer
SLG is grown on Cu foil (35 μm thick) at 1050 °C via low-pressure chemical vapor deposition (CVD), employing a quartz tube furnace. SLG on Cu is then wet transferred onto the target substrates[65]: A4-950K ply(methyl-methacrylate) polymer (PMMA) is spin coated at 2000 rpm on the top surface of the sample (1 cm × 1 cm), followed by 1 min baking on a hot plate at 90 °C. Mild oxygen plasma is utilized to remove SLG on the bottom surface. After that, the PMMA/SLG/Cu sample is placed in a solution of 1 g of ammonium persulfate and 40 ml of DI water to etch Cu. Once Cu etching is complete, the PMMA-SLG film is transferred in a beaker with DI water and then lifted with the previously prepared substrate hosting the CSRR array. This sample is left to dry overnight, followed by PMMA removal with acetone.

### FEM simulations
The CSRR single unit element with the geometry illustrated in Fig. 1b comprises a perfect conductor C-shaped ribbon, embedded at the interface between a 300-nm-SiO$_2$/Si bottom and air top domains. The unit cell pitch is $p = 14.5$ μm, and the total height of the calculation domain is 85 μm, accounting for 35 μm dielectric and 50 μm air. The FEM calculations (by COMSOL Multiphysics) are carried out by setting Floquet boundary conditions along the x,y side edges, and scattering boundary conditions on the top and bottom domain edges (z edges). The S-parameters are extracted by placing one input port on top of the air domain and one receiver port on the bottom of the Si domain. The transmittance is calculated by the square magnitude of the S$_{21}$

parameter, $T = |S_{21}|^2$. A$_G$ is derived as the ratio between electric field magnitude averaged over the gap region, defining a box centered in the gap of volume $5 \times 5 \times 5$ μm$^3$, and the field magnitude averaged over a box volume of identical size, positioned at the center of the ring (no-enhancement region).

### Device fabrication
The SLG CSRR is fabricated on a high ($10^4$ Ωcm)-resistivity 300-μm-thick Si substrate to ensure THz transparency, coated with 300 nm SiO$_2$ (by Siltronyx). The CSRR array pattern is defined by optical lithography using a LOR3A/S1805 bilayer photoresist, on a $6 \times 6$ mm$^2$ area, followed by metal evaporation and liftoff of 10 nm/80 nm of Cr/Au. SLG is then transferred on the top of the CSRR array, via a PMMA assisted method[65], using a $1 \times 1$ cm$^2$ film. Then, the SLG in the gap area is defined with a second optical lithography step for SLG mask photoresist pattering, followed by a plasma-O$_2$ etching to remove the SLG film from the desired area, and a final cleaning by acetone soaking. The single-plasmon THz QCL, emitting at 3.21 THz (λ = 107 μm), employed for the pump and probe experiments, exploits a bound to continuum-optical phonon hybrid active region design. The 25-μm-thick GaAa/AlGaAs active region is patterned into a surface-plasmon waveguide onto a GaAs substrate through a combination of optical lithography and metal deposition. A 5-nm-thick, 40-μm-wide Ni side-absorber is introduced on each edge of the ridge to increase the difference in losses between fundamental and higher order transverse modes, to fully suppress the higher-order competing modes. An overlap of 3 μm between each Ni side-absorber and the upper Au over-layer, 150 nm thick, is set by design. The 700-nm-thick heavily Si-doped ($5 \times 10^{18}$ cm$^{-3}$) GaAs top contact layer lies between active region and substrate.

### Scattering scanning near-field optical microscopy
The THz QCL is operated in CW, with radiation collimated by a 50 mm focal length parabolic mirror, focused by two parabolic mirrors (f/1 = 2 inches) on an AFM tip (25PtIr300BH-40, 80 μm long Rocky Mountain Nanotechnology, 40 nm apex-radius) of an s-SNOM microscope (Neaspec/Attocube). The THz radiation is concentrated at the tip apex by the lightning-rod effect and adiabatic compression. The radiation backscattered by the tip (-10$^{-3}$ of the emitted power) is then coupled into the laser cavity and the SM signal detected as a perturbation (ΔV) to the QCL contact voltage. The AFM tip is operated in tapping mode at a frequency Ω-22 kHz, such that the near-field contribution to the SM signal can be isolated by lock-in detection (with a UHFLI from Zurich Instruments) at the tapping frequency harmonics nΩ, with $n = 2$, 3, 4, and 5.

## Data availability
The data presented in this study are available on request from the corresponding author.

## Code availability
The relevant computer codes supporting this study are available from the authors upon request.

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

## Acknowledgements

We thank Oleg Mitrofanov for helpful discussions. We acknowledged funding from the European Union through the FET Open project EXTREME IR (944735), the Graphene Flagship, and EPSRC programme grants HyperTerahertz (EP/P021859/1) and TeraCom (EP/W028921/1), EPSRC Grants EP/K01711X/1, EP/K017144/1, EP/N010345/1, EP/L016087/1, EP/X015742/1, EP/V000055/1, EP/X015742/1, ERC grants Hetero2D, GSYNCOR, GIPT, EU grants CHARM, Graph-X.

## Author contributions

M.S.V. conceived and supervised the experiment. A.D. fabricated the devices, set up the transport and optical experiment, performed numerical simulations. M.S.V. and A.D. interpreted the data. C. Song and S.S.D., designed the and performed preliminary optical measurements of the CSRR. C.Schiattarella. performed the s-SNOM characterization. L.H.L, M.S, A.G.D. and E.H.L grew by molecular beam epitaxy and optimized the QCL structure. J.Z., O.B., and A.C.F grew the high crystalline quality graphene film on Cu and conducted Raman characterization. The manuscript was written by M.S.V. and A.D. All authors contributed to the final version of the manuscript and to discuss the results.

## Competing interests

The authors declare no competing interests.
