## [Peer Review File · Nature Communications]

Compact terahertz harmonic generation in the Reststrahlenband using a graphene-embedded metallic split ring resonator arrayEditorial Note: This manuscript has been previously reviewed at another journal that is not operating a transparent peer review scheme. This document only contains reviewer comments and rebuttal letters for versions considered at *Nature Communications*.

REVIEWER COMMENTS

Reviewer #2 (Remarks to the Author):

My previous comments on the manuscript “Harmonic generation in the Reststrahlenband using graphene-embedded metallic split ring array, optically pumped by a semiconductor heterostructure laser” by Gaspare et al. have been properly addressed. In the new version of the manuscript, the experimental results are convincingly explained and support the main claims. The main achievement of the manuscript, in my opinion, is the combination of the previous results of efficient THz THG generation using high power laser systems with metamaterial THz field enhancement. Using such a combination, Gaspare et al. demonstrated THz THG at about 9 THz frequency using CW sources. Although the efficiency of the THz THG in the manuscript is low, I find the results interesting and important for the further development of THz photonics based on 2D systems. Therefore, I recommend the manuscript for publication in Nature Communications after minor revisions.

1. In the supplementary Information it is still written “Reststrahlenband”.

2. Lines 57-58: Spintronic emitters can be also employed to generate broad-band THz light in a pumped optical scheme via femtosecond excitation²⁰ and are based on down-conversion. If spintronic emitter is pumped by femtosecond laser, then the down-conversion process will take place. But if someone pumps spintronic emitter with THz light, then up-conversion will take place (Nature Communications 14, 7010, 2023).

3. Lines 88-91: The field enhancement in the split gap of each CSRR^{36,37}, one order of magnitude larger than the absorption in un-patterned film, leads to the formation of an intense hotspot in the active film^{38,39}, confirmed both by electromagnetic simulations and near-field nanoimaging, which determines an out-of-equilibrium hot electron state^{28,40}. There are many very long sentences in the manuscript that are very difficult to read and understand (like this one). The authors should describe how to make a direct comparison between two different phenomena (field enhancement is larger than absorption).

4. Lines 134-137: Our CSRR arrays has resonant modes at frequencies higher than the fundamental, at the third harmonic frequency (Fig.1f), boosting the overall THG efficiency (THGE), defined as the variation of optical Transmittance at the third harmonic frequency, respect to the linear response at the fundamental frequency, normalized over the linear response²⁵. It is very difficult to understand the meaning of this sentence. Does it mean that THGE is determined as a ratio between nonlinear transmission changes and linear transmission? Or is some additional normalization required? What does linear response mean here? The authors should describe more clearly why the additional resonance of the CSSR at the third harmonic frequency leads to the enhancement of THGE.

5. Lines 242-245: Under steady state excitation⁴⁵, where the optical beam is either continuous wave or

with a pulse duration much longer than the SLG cooling time (> 2.5 ps) 28,46,47, the impinging electric field induces a polarization current in the SLG, oscillating in the same direction of the incoming field (see SI, Section 5) at the frequency of the incident wave ν (Kerr effect).

The Kerr effect is typically associated with a third order nonlinear process involving the conjugation of two oscillating fields: “ $+\Omega+\Omega-\Omega=\Omega$ ” (as described in Ref 27). In this case, the nonlinear polarization current will indeed oscillate at the Ω frequency, and it will be collinear with the polarization of the fundamental radiation. However, the effect when polarization currents oscillate in-takt with fundamental radiation in conducting materials only due to Kerr effect needs to be clarified. Drude response still should be much larger than the Kerr effect, and the electric current dynamics should be dominated by Drude response, while the Kerr effect is a perturbation.

6. Lines 350-363: The authors described here the absences of the even-order nonlinear response of metamaterials at THz frequencies. Recent manuscript (<https://doi.org/10.1093/nsr/nwad136>) needs to be mentioned here, as second order nonlinear dynamics at THz frequencies have been demonstrated in split ring resonators.

Reviewer #2 (Remarks to the Author):

My previous comments on the manuscript “Harmonic generation in the Reststrahlenband using graphene-embedded metallic split ring array, optically pumped by a semiconductor heterostructure laser” by Gaspare et al. have been properly addressed. In the new version of the manuscript, the experimental results are convincingly explained and support the main claims. The main achievement of the manuscript, in my opinion, is the combination of the previous results of efficient THz THG generation using high power laser systems with metamaterial THz field enhancement. Using such a combination, Gaspare et al. demonstrated THz THG at about 9 THz frequency using CW sources. Although the efficiency of the THz THG in the manuscript is low, I find the results interesting and important for the further development of THz photonics based on 2D systems. Therefore, I recommend the manuscript for publication in Nature Communications after minor revisions.

ANSWER

We thank the reviewer for the positive evaluation of our revised manuscript and for recommending it for publication in *Nature Communications*

1. In the supplementary Information it is still written “Reststrahlenband”.

ANSWER

We thank the reviewer for pointing this out. We corrected the remaining misspelling in the Supplementary Information.

2. Lines 57-58: Spintronic emitters can be also employed to generate broad-band THz light in a pumped optical scheme via femtosecond excitation,²⁰ and are based on down-conversion. If spintronic emitter is pumped by femtosecond laser, then the down-conversion process will take place. But if someone pumps spintronic emitter with THz light, then up-conversion will take place (Nature Communications 14, 7010, 2023).

ANSWER

We thank the referee on his/her remark about the possibility to exploit spintronic emitters for both up- and down-conversion. The experiment reported in the mentioned paper on up-conversion has now been added as ref. 20 in the revised manuscript version.

However, we wish to point out that this system employs again fs-laser-based and accelerator-based sources. In our system we use, as a pump, a compact, electrically driven, quasi-CW THz QCL, that possesses an output power high enough, when coupled to graphene and resonators, for photon generation in the Reststrahlenband band. Spintronic heterostructures could potentially be an interesting avenue to explore in the future.

To address the referee remark, we modified the revised text as follows.

Page 2, line 17

Alternative commercial systems include time-domain spectroscopy (TDS) in which III-V photoconductive switches¹⁷, or more recently spintronic heterostructures,^{18,19} generate coherent radiation pulses through femtosecond excitation, but these are also spectrally broad. Furthermore, TDS systems are not compact and emit most power over the ~ 0.2 – 4 THz range. Spintronic emitters can be also employed to generate broad-band THz light in a pumped optical scheme via up-conversion²⁰.

and we added the references:

19. Dang, T. H. et al. Ultrafast spin-currents and charge conversion at 3d-5d interfaces probed by time-domain terahertz spectroscopy. Appl. Phys. Rev. 7, 41409 (2020).

20. Ilyakov, I., Brataas, A., de Oliveira, T.V.A.G. et al. Efficient ultrafast field-driven spin current generation for spintronic terahertz frequency conversion. Nat Commun 14, 7010 (2023). <https://doi.org/10.1038/s41467-023-42845-8>

3. Lines 88-91: The field enhancement in the split gap of each CSRR^{36,37}, one order of magnitude larger than the absorption in un-patterned film, leads to the formation of an intense hotspot in the active film^{38,39}, confirmed both by electromagnetic simulations and near-field nanoimaging, which determines an out-of-equilibrium hot electron state^{28,40}.

There are many very long sentences in the manuscript that are very difficult to read and understand (like this one). The authors should describe how to make a direct comparison between two different phenomena (field enhancement is larger than absorption).

ANSWER

In the unpatterned film (i.e. when no resonator is defined), the nonlinear response in graphene arises from the direct absorption of the graphene film. This is the case presented in the seminal work of Hafez et al., (ref. 25 in our manuscript), showing the first experimental demonstration of high harmonic generation in graphene using a free electron laser. The introduction of a metamaterial-based structures, such as grating ribbons (see ref. 32 in the manuscript), micro-disks (ref. 30 in the manuscript), or circular split ring resonators (present work, amongst others), has proven to be a very proficient way to concentrate the impinging electric field in the film. This allows reducing the power requirements for harmonic generation, overcoming the limitations of the low-absorption of atomically thin single layer graphene.

In the SRR, this led to the formation of a hot-spot in the split gap, which is clearly not possible in the pristine film. The quantitative comparison between the un-patterned film and the resonator-coupled film is then possible. We consider the power density from the focused, free-space beam that impinges on the bare graphene surface, a parameter that can be determined accurately from the measured power and focus spot size, and the power density in the field enhancement area that is calculated through the electromagnetic simulations described in the main article at page 5.

To address the referee's remark, we rephrased the text as follows and have gone through the article to reduce the length of the sentences

Page 4, line 6

The field enhancement in the split gap of each CSRR^{37,38}, which is one order of magnitude larger than the absorption in an un-patterned film (i.e. no resonator)^{39,40}. In the un-patterned graphene, the nonlinear response arises from the direct absorption of the graphene film²⁶. The introduction of a metamaterial-based structure is an effective solution to concentrate the optical field in the graphene layer^{32,30}, overcoming the limitations of the low-absorption of atomically thin single layer graphene. In the CSRR case, this led to the formation of a hot-spot in the split gap which determines an out-of-equilibrium hot electron state^{29,41}, as confirmed both by electromagnetic simulations and near-field nano-imaging (see below).

4. Lines 134-137: Our CSRR arrays has resonant modes at frequencies higher than the fundamental, at the third harmonic frequency (Fig.1f), boosting the overall THG efficiency (THGE), defined as the variation of optical Transmittance at the third harmonic frequency, respect to the linear response at the fundamental frequency, normalized over the linear response²⁵.

It is very difficult to understand the meaning of this sentence. Does it mean that THGE is determined as a ratio between nonlinear transmission changes and linear transmission? Or is

some additional normalization required? What does linear response mean here? The authors should describe more clearly why the additional resonance of the CSSR at the third harmonic frequency leads to the enhancement of THGE.

ANSWER

We employed the definition of THG efficiency used in previous works. When the system nonlinear response is stimulated, the detected optical signal deviates from the expected linear signal. The THG efficiency is then defined as the signal intensity variation at 3 times the fundamental frequency ν_0 , over the signal intensity at the fundamental frequency, here named as the linear response.

In the present case, the laser beam illuminating the SRR array, would be the transmitted intensity to the detector after interacting with the SRR sample.

This total recorded signal comprises both the linearly transmitted signal and the THG component. The experimental configuration, illustrated in Fig. 4a allows us to isolate, spectrally, only the THG term using the optical filter, as the only recorded signal is the emission from the SRR array at the third harmonic. This signal is then normalized by the total signal recorded without the filter, i.e. the intensity of the pump beam, where the much smaller THG component of the total signal can be neglected. Such a normalization allows then to perform the direct quantitative comparison of the THG efficiency with the model. The latter was calculated by combining the fundamental and third harmonic terms separately, following the method described in the main article at pages 11-13.

To better clarify this point, we modified the text as follows:

Page 12 Line 13

When the system nonlinear response is stimulated, the detected optical signal deviates from the expected linear signal. The laser beam illuminating the SRR array is transmitted to the detector after interacting with the SRR sample. The nonlinear response induces a transmittance variation $\Delta T = (T_0 - T_3)/T_0$, whose frequency dependence (Fig. 3e) can be derived from the linear (T_0), at and non-linear (T_3) transmittances as:

$$[T_0, T_3(\nu)] = \frac{4n_{sub}}{Z_0^2 \left| \frac{n_{sub}}{Z_0} + [\sigma_0, \sigma_{3,tot}(\nu)] \right|^2} \quad (3)$$

where Z_0 is the vacuum impedance. Such parameter also represents the THG efficiency, defined as the signal intensity variation at $3\nu_0$ over the signal intensity at the fundamental (linear) frequency.

The THG experiment conceived in our work comprises the optical excitation of an array of micro resonators, whose fundamental frequency is tuned to match the pump source. The pump signal is absorbed by the graphene-embedded resonator, driving and enhancing its non-linear response. This is then up-converted and emitted at the $3\nu_0$ frequency. While it is straightforward to envision the micro-resonator role in field enhancement of the pump in the up-conversion of the incoming field, the optical re-emission of the THG signal is also enhanced by the micro-resonator. Indeed, the numerical calculations show that the CSRR also possesses resonances at the TH, whose field profile is similar to the harmonic mode, i.e. with the field intensity in the split gap. This boosts its functionality as a TH emitter.

To better clarify this point, we added the following sentence:

Page 6, line 8

The optimal combination of gap size, enhancement factor and resonance frequency, is shown for a 1.5- μm split-gap. This achieves a maximum enhancement factor ~ 13 , corresponding to a calculated quality factor $Q \sim 13$ for a CSRR designed to be resonant at 3.21 THz. In the devised experiment, the pump signal is absorbed by the graphene-embedded resonator, driving, and enhancing its non-linear

response. The pump is then up-converted for generation at $3\nu_0$. While it is straightforward to envision the CSSR role in the field enhancement of the pump in the up-conversion of the incoming field, the CSSR has also a role in the emitted third harmonic signal. Indeed, the CSSR arrays has resonant modes at the third harmonic frequency (Fig.1f), hence boosting the overall THG efficiency.

5.Lines 242-245: Under steady state excitation⁴⁵, where the optical beam is either continuous wave or with a pulse duration much longer than the SLG cooling time (> 2.5 ps) ^{28,46,47}, the impinging electric field induces a polarization current in the SLG, oscillating in the same direction of the incoming field (see SI, Section 5) at the frequency of the incident wave ν (Kerr effect).

The Kerr effect is typically associated with a third order nonlinear process involving the conjugation of two oscillating fields: “ $+\Omega+\Omega-\Omega=\Omega$ ” (as described in Ref 27). In this case, the nonlinear polarization current will indeed oscillate at the Ω frequency, and it will be collinear with the polarization of the fundamental radiation. However, the effect when polarization currents oscillate in-takt with fundamental radiation in conducting materials only due to Kerr effect needs to be clarified. Drude response still should be much larger than the Kerr effect, and the electric current dynamics should be dominated by Drude response, while the Kerr effect is a perturbation.

ANSWER

We would like to remark that we mention the Kerr response only to account for the 3rd order non-linear response. The remark of the referee is correct and the linear response at the fundamental frequency is fully described by the Drude model in graphene.

According to many experimental reports (see refs 25,26,27, 30-33 of the revised manuscript), the non-linear response in graphene relies on the hot carrier population driven by efficient intraband absorption of an intense optical beam. Such response can be accurately modeled by using the Drude model for graphene, considering the heating of the carrier population.

In this condition, the polarization current is related only to the hot carrier heating/cooling dynamics, and to the time profile of the excitation beam. However, at higher optical power, the presence of inherent nonlinear components in the optical response (such as the Kerr term) also suggests the presence of the field-driven nonlinearity beyond thermal effects. Thus, we invoke the Kerr effect to account for the presence of field-dependent, higher order terms in the optical conductivity, that appear to lead to the observation of other nonlinearities that depend on the strength of the driving electric field, and are responsible for the THG signal observed here. Besides, we point out that the Drude term was always included in our model, by considering the T_e -dependence in the intraband conductivity of eq.2.

A quantitative comparison of the different weight of the Drude-like and Field-driven terms in building up the graphene nonlinear response has been presented in our recent paper *ACS Photonics* 2023, 10, 9, 3171–3180 (now ref. 42 in the revised version). This shows that in the un-patterned graphene film the Drude thermal effect dominates over field-driven effects for moderate input power density/doping levels, as correctly remarked by the referee. Conversely, when accounting for the field enhancement of the SRR, the field-driven third-order nonlinearity becomes the dominating term in establishing the THG efficiency, as visible from the following plot showing the comparison between the THG efficiency calculated considering only the Kerr 3rd order term (blue curve), only the Drude hot-electron term (black curve), or the combination of both terms (red trace).

To better clarify this point, we modified the text as follows:

Page 11, line 3

Under steady state excitation⁴⁶, where the optical beam is either continuous wave or with a pulse duration much longer than the SLG cooling time (> 2.5 ps)^{29,47,48}, the impinging electric field induces a polarization current in the SLG, oscillating in the same direction of the incoming field (see SI, Section 5) at the frequency of the incident wave.^{49,27,28} The non-linear response in graphene relies on the hot carrier population driven by efficient intraband absorption. It can be accurately modeled by using the Drude response for graphene, taking into account the heating/cooling dynamics of the carrier population, and the time profile of the excitation beam^{25,30,42}. However, at higher optical field intensity (≥ 10 kV/cm), the presence of inherent nonlinear components in the optical response also suggests the presence of the field-driven nonlinearity beyond thermal effects, accounted by the Kerr term⁴² (see Supplementary Information).

And we added the paragraph to section S5.2 to the Supplementary Information

We use the Kerr effect to calculate the field-dependent, higher order terms in the optical conductivity, that is the main responsible of the observation of non-linear effects depending on the strength of the driving electric field, responsible of the observed THG signal. Furthermore, in eq. S7, the T_e -dependence was also considered in the intraband term, namely:

$$\sigma_{\text{intra}}(\nu) = \frac{-iD_{\text{he}}}{\pi} \frac{1}{(2\pi\nu + i\Gamma_{\text{he}})} \quad (\text{S12})$$

A quantitative comparison of the different weight of the Drude-like and field-driven terms in building up the graphene nonlinear response has been presented in our recent paper²³. In Ref. 23 it is shown that in the unpatterned film, the Drude thermal effect dominated over field-driven effects for moderate input power density/doping levels. Conversely, when accounting for the field enhancement of the SRR,

the field-driven third-order nonlinearity becomes the dominating term in establishing the THG efficiency, as visible from Figure S9, showing the comparison between the THG efficiency, calculated considering only the Kerr 3rd order term (blue curve), only the Drude hot-electron term (black curve), or the combination of both terms (red trace).

Figure S9: $\Delta T/T_0$ ($3\nu_0$), i.e. THG efficiency, as a function of frequency in the pumped SLG-CSRR, calculated considering the 3rd order Kerr term (blue line), the hot-electron Drude like response (black line), or the combination of both terms (red trace).

With the added ref.S23

23. Di Gaspare, A. *et al.* Electrically Tunable Nonlinearity at 3.2 Terahertz in Single-Layer Graphene. *ACS Photonics* (2023) doi:10.1021/acsp Photonics.3c00543.

6. Lines 350-363: The authors described here the absence of the even-order nonlinear response of metamaterials at THz frequencies. Recent manuscript (<https://doi.org/10.1093/nsr/nwad136>) needs to be mentioned here, as second order nonlinear dynamics at THz frequencies have been demonstrated in split ring resonators.

ANSWER

In the recent interesting paper mentioned by the referee, the authors demonstrate that a combination of impact ionization and non-linear Thomson scattering permits SHG using SRRs, with a design concept similar to the one used in our work, and realized on a centrosymmetric nonlinear Si wafer. The proposed paradigm has great potential to realize even-order harmonic generation in many material platforms ruled out by their centrosymmetric nature; however, it requires a mechanism such as the impact ionization in solids, to multiply and accelerate the free carriers and establish a significant out-of-plane Lorentz force. While the reported field enhancement from the SRR (field enhancement 15) is in broad agreement with the value of our design (field enhancement 13) the SHG was reported exploiting a tens of kV/cm

pump energy, for driving the carrier to the impact ionization, to be compared with $\lesssim 1\text{kV/cm}$ of our work, in an effective volume corresponding to 5 micron-thick doped silicon. Although the impact ionization is virtually possible in graphene (see for example *J. Appl. Phys.* 112, 093707 (2012)), the carrier density ($\gtrsim 10^{18}\text{ cm}^{-3}$)/field-driven velocity (field $\gtrsim 25\text{kV/cm}$) thresholds needed to match the conditions of the experiment of Y. Wen and coworkers (Ref. 59) that are orders of magnitude higher than the current experimental condition.

Furthermore, the electromagnetic response of the CSRR is dominated by the magnetic coupling, resulting from the induction current in the circular ring geometry. However, the magnetic coupling enhancement of the CSRR design is spatially separated from the electric field area, as shown in the following image of the 2D field distribution, at resonance, of the electric (left) and magnetic fields (right). As we explained in our work, our rationale was to embed the graphene film in the split gap area, to exploit the electric field enhancement that can drive THG in graphene.

The effect invoked for the SHG in SRR would not be effective in our sample because the active film and the coupling area did not overlap. Indeed, no signatures of SHG were observed in our experiment. Of course, this is a potentially interesting avenue to investigate in future studies.

Based on the above discussion, we can conclude that the observation of the SHG prompted by 2nd order Thomson scattering does not seem possible in our current geometry.

To address the referee's remark we modified the text as follows:

Page 16, line 12

Although the introduction of non-centro-symmetric resonator arrays could activate/enhance the even-order nonlinearities^{59,60}, the efficiency of this effect is likely too weak in the proposed system³². In ref.⁶⁰, the reported SHG in CSRRs relies on a combination of impact ionization and non-linear Thomson scattering, to multiply and accelerate the free carriers and establish a significant out-of-plane Lorentz force⁶⁰. While the field enhancement from the asymmetrical SRR ($A_G \sim 15$)⁶⁰ is in broad agreement with the value of our design ($A_G \sim 13$), SHG was observed with a tens of kV/cm pump field in a FEL facility, to drive the carrier to the impact ionization. Conversely, we here adopt fields $\lesssim 1\text{kV/cm}$. Although the impact ionization is virtually possible in graphene⁶¹, the carrier density ($\gtrsim 10^{18}\text{ cm}^{-3}$)/field-driven velocity (field $\gtrsim 25\text{kV/cm}$) thresholds are orders of magnitude higher than the current experimental condition. Furthermore, in the employed CSRR structure, with the graphene patterned only in the split gap, the magnetic coupling enhancement of the CSRR design is spatially separated from the electric field area (see Supplementary Information). In the present experiments, high harmonic generation results only from the electric field enhancement, dominant at both the fundamental and at the third harmonic frequencies. At the second harmonic, on the other hand, the same amplification was not retrieved.

With the added references

60. Wen, Y. *et al.* A universal route to efficient non-linear response via Thomson scattering in linear solids. *Natl. Sci. Rev.* **10**, nwad136 (2023).
61. Pirro, L., Girdhar, A., Leblebici, Y. & Leburton, J.-P. Impact ionization and carrier multiplication in graphene. *J. Appl. Phys.* **112**, 93707 (2012).

and we added section S8 to the Supplementary Information.

S8: Electro-Magnetic Coupling in CSRR

The electromagnetic response of the CSRR is dominated by the magnetic coupling, resulting from the induction current in the circular ring geometry. However, the magnetic coupling enhancement region of the CSRR design is spatially separated from the electric field area, as shown in Figs. S12 plotting the bi-dimensional distribution, at resonance, of the electric (Fig. S12a) and magnetic fields (Fig. S12b). Embedding graphene in the split gap area, allows exploiting the electric field enhancement to drive an efficient THG.

Figure S12: Bi-dimensional field distribution of the CSRR at the resonance, for the electric (a) and magnetic (b) field.

In the following, we address the additional comments on the concerns raised by referee #1 during the previous round of review at Nature Photonics, listed in a separate document attached to the review.

I find most of the responses to reviewer#1's comments adequately addressed. At the same time, a few points need clarification:

Answer on page 1:

b) The referee is correct that the use of spintronic emitters has been proven to be a successful approach to generate broad-band THz light with a femtosecond optical pumped scheme, i.e. employing a fiber laser. The generation mechanism, exploiting the net photocurrent arising from the femtosecond excitation of a spin population in a magnetic material, leads to the generation of spectral broad short THz pulses, and the core principle is based on down-conversion of an optical/near-infrared beam. We have added a reference to highlight this approach (see below)

*However, we wish to point out that spintronic emitters can be only used with femtosecond excitation, leads to broad-band emission and are based on down-conversion. Further the efficiency reduces with THz frequency in these structures. In the present case, **no femtosecond laser is used and we are upconverting the QCL emission.** Although, as highlighted by the referee, the spectral coverage would be well within the Reststrahlenband of III-V materials and beyond, as far as we know, **no demonstration of CW emission so far has been reported employing spintronic systems.***

This answer is not exactly correct. When spintronic emitters are illuminated with THz pulse, the THz frequency multiplication (up-conversion) can take place. The authors need to consider recent results (Nature Communications 14, 7010, 2023) or remove part of the text about the limitations of the spintronic emitters due to the use of femtosecond lasers.

ANSWER

We thank the referee on his/her remark about the possibility to exploit spintronic emitters for both up- and down-conversion. The experiment reported in the mentioned paper on up-conversion has now been added as ref. 19 in the revised manuscript version.

However, we wish to point out that this system employs again fs-laser-based and accelerator-based sources. In our system we use, as a pump, a compact, electrically driven, quasi-CW THz QCL, that possesses an output power high enough, when coupled to graphene and resonators, for photon generation in the Reststrahlen band. Spintronic heterostructures could potentially be an interesting avenue to explore in the future.

To address the referee remark, we modified the revised text as follows.

Page 2, line 17

Alternative commercial systems include time-domain spectroscopy (TDS) in which III-V photoconductive switches¹⁷, or more recently spintronic heterostructures,^{18,19} generate coherent radiation pulses through femtosecond excitation, but these are also spectrally broad. Furthermore, TDS systems are not compact and emit most power over the $\sim 0.2 - 4$ THz range. Spintronic emitters can be also employed to generate broad-band THz light in a pumped optical scheme via up-conversion²⁰.

And we added the references

19. Dang, T. H. et al. Ultrafast spin-currents and charge conversion at 3d-5d interfaces probed by time-domain terahertz spectroscopy. Appl. Phys. Rev. 7, 41409 (2020).

20. Ilyakov, I., Brataas, A., de Oliveira, T.V.A.G. et al. Efficient ultrafast field-driven spin current generation for spintronic terahertz frequency conversion. Nat Commun 14, 7010 (2023). <https://doi.org/10.1038/s41467-023-42845-8>

Answer on page 5:

To better clarify this point, we added the following sentence in the revised manuscript:

Page 5, Line 8

Hence, the CSRR design offers the unique possibility to exploit the split gap, outside the metal region, for graphene embedding, to concentrate the peak electric field in a portion of the active material where no screening or absorption from the metallic counterpart of the array dominates, differently from the simpler grating arrays³¹, whose electromagnetic coupling efficiency drops as $1/f$ ³², hence being not suitable for the high THz frequencies aimed in the present work.

It would be much easier to read it if such a long sentence was broken up into fewer sentences.

ANSWER

To address the referee' comment we rephrased as follows

Page 4 line 22

Hence, the CSRR design offers the capability to exploit the split gap for graphene embedding. This allows to concentrate the peak electric field in a portion of the active material where no screening or absorption from the metallic counterpart of the array dominates. This is in contrast to the simpler grating arrays³², where the electromagnetic coupling efficiency drops as $1/f$,³³ hence being less suitable for the high THz frequencies aimed in the present work.

Answer on page 6:

ANSWER

It is a combination of both. The SRR possesses both e.m. type response: the inductive current in the ring and the dipolar displacement in the split gap. Usually, the electromagnetic response of this class of resonators is described as mainly magnetic (see for example ref. 34 of the main article). For the targeted application, we are exploiting the dipolar field in the split gap, generated as a consequence of the ring induction current.

To better clarify this point we added the following sentence to the revised manuscript:

Page 7, line 16:

The electromagnetic coupling mechanism between the SRR and graphene is a combination of the capacitive coupling due to the gap and the inductive coupling due to the ring.

Such an answer is too broad and does not cover the particular case presented in the manuscript – when the graphene region is located in the gap. In the SRR, the magnetic and electric field enhancement (and couplings) are spatially separated. If the main mechanism is based on capacitive or inductive coupling in the gap region– is not clear. It would be good if the authors could provide some references or show simulations to address this question in more detail.

ANSWER

The referee' observation of the separation between the magnetic and electrical coupling is correct, (see our reply to point 6 in the present letter). The magnetic inductive current helps to build up the electric field in the split gap, which is the actual mechanism to enhance the field for THG and permits a SRR with a high Q -factor. This is shown in the figure reported below, that plots the maps of the electric field obtained in our design and in a split-gap ribbon design equivalent to the SRR, i.e. having the same metal width, split gap and with the same resonance frequency. The field enhancement at the resonance in the split gap is visible in both designs, with comparable values (13 in the SRR and 18 in the split ribbon resonator), but the Q -factor in the SRR is significantly larger ($Q_{\text{SRR}}=13$, $Q_{\text{ribbon}}=1.8$). Besides, the optimal geometry of the CSRR allows the realization of a much dense array, a property which is more desirable for THG efficiency.

Following the referee's remark, we rephrased the main text as follows:

Page 5 Line 3

The electromagnetic coupling mechanism between the CSRR and graphene is a combination of the capacitive coupling due to the CSRR gap and the inductive coupling due to its ring shape. The magnetic inductive current helps to build up the electric field in the split gap, which is the devised mechanism adopted to enhance the field for THG.

Page 6 Line 25

If compared to linear grating dipolar resonators, the ring design permits a higher Q -factor (See Supplementary Information).

and we added the following section in the **Supporting Information** file, section S8:

The electric field in the split gap, is by-design, the mechanism responsible for the enhancement of the field for THG. The CSRR design permits a resonance with a higher Q -factor. This is shown in figure S13, that plots the maps of the electric field distribution in the devised CSRR and in a split-gap ribbon having the same metal width, split gap and with the same resonance frequency of our CSRR. The field enhancement at the resonance in the split gap is visible in both designs, with comparable values (13 in the SRR and 18 in the split ribbon resonator), but the Q -factor in the SRR is significantly larger ($Q_{SRR}=13$, $Q_{ribbon}=1.8$). Besides, the optimal geometry of the CSRR allows the realization of a denser array, which is more desirable for a more efficient THG process.

Figure S13: a,b) Bi-dimensional profile of the electric field at resonance ($\nu_0 = 3.21\text{THz}$) obtained in the CSRR employed in this work (a), and in an equivalent linear split-gap ribbon (b), whose geometry was designed to have

the same split-gap, metal ribbon width and resonant frequency of a). c) Comparison of the calculated transmittance for the CSRR in a) (black curve) and split-gap ribbon (red curve).

Answer on page 7:

Page 15, line 3

It is worth mentioning that no visible signal at the SH frequency is retrieved. In fact, a SHG signal would rely on a nonzero second-order nonlinearity ($\chi^{(2)} \neq 0$), which is prohibited by the purely centro-symmetrical nature of the hexagonal, atomically-thin, graphene structure. An anisotropic SHG signal could in principle arise in layered graphene on SiO₂^{57,58}, if the crystal orientation of the graphene hexagonal structure with respect to the substrate crystal fulfills the conditions for dipolar symmetry breaking. However, in the polycrystalline, single-layer film used in the present work, comprising single crystalline domains of few micron size and random crystalline orientation, the latter condition is not fulfilled. Although the introduction of non centro-symmetric resonator arrays could activate/enhance the even-order nonlinearities⁵⁹, the efficiency of this effect in the intraband THz absorption is likely too weak and it has not been reported so far³¹. Furthermore, the employed CSRR structure, with the graphene patterned only in the split gap, is only weakly asymmetrical. High harmonic generation in the devised system relies on the resonator field enhancement, significant at both the fundamental and at the TH frequencies. At the SH, on the other hand, the same amplification was not retrieved.

I find this answer not complete, and it is also related to the previous answer about coupling. In the case of SRR, the symmetry is broken, and THz SHG can be realized (<https://doi.org/10.1093/nsr/nwad136>). Such effects have been studied some time ago also in different spectral ranges. The authors need to describe if they did not observe THz SHG either due to low sensitivity (SHG could be much weaker than THG), or in the gap the inductive coupling is not sufficient to initiate second order effects.

ANSWER

We thank the reviewer for this remark.

The reply to this comment is reported above in reply to comment 6.

This is an interesting point, and the inductive coupling could in principle introduce second-order Thomson scattering. The key elements preventing the observation of the SHG driven by this mechanism is the carrier multiplication driving force, that in the paper of *Y. Wen et al.* is delivered by a very high intensity pump field, and which is combined with the same split gap field enhancement we have in our SRR design. Furthermore, the region of the magnetic force coupling in the SRR, which appears to activate the SHG, is separated from the graphene embedded area, the split gap, in our sample.

However, this could certainly be an interesting research avenue in the future with an optimized design.

REVIEWERS' COMMENTS

Reviewer #2 (Remarks to the Author):

All my comments have been properly addressed. I therefore recommend the manuscript for publication in Nature Communications.